# *Porphyromonas gingivalis* induction of TLR2 association with Vinculin enables PI3K activation and immune evasion

**Karthikeyan Pandi**[1], **Sarah Angabo**[1], **Jeba Gnanasekaran**[1], **Hasnaa Makkawi**[1], **Luba Eli-Berchoer**[1], **Fabian Glaser**[2], **Gabriel Nussbaum**[1]*

**1** Institute of Biomedical and Oral Research, Hebrew University-Hadassah Faculty of Dental Medicine, Jerusalem, Israel, **2** Bioinformatics Knowledge Unit, The Lorry I. Lokey Interdisciplinary Center for Life Sciences and Engineering, Technion—Israel Institute of Technology, Haifa, Israel

* gabrieln@ekmd.huji.ac.il

**Data Availability Statement:** This study includes no data deposited in external repositories. All

## Abstract

*Porphyromonas gingivalis* is a Gram-negative anaerobic bacterium that thrives in the inflamed environment of the gingival crevice, and is strongly associated with periodontal disease. The host response to *P. gingivalis* requires TLR2, however *P. gingivalis* benefits from TLR2-driven signaling via activation of PI3K. We studied TLR2 protein-protein interactions induced in response to *P. gingivalis*, and identified an interaction between TLR2 and the cytoskeletal protein vinculin (VCL), confirmed using a split-ubiquitin system. Computational modeling predicted critical TLR2 residues governing the physical association with VCL, and mutagenesis of interface residues W684 and F719, abrogated the TLR2-VCL interaction. In macrophages, VCL knock-down led to increased cytokine production, and enhanced PI3K signaling in response to *P. gingivalis* infection, effects that correlated with increased intracellular bacterial survival. Mechanistically, VCL suppressed TLR2 activation of PI3K by associating with its substrate PIP2. *P. gingivalis* induction of TLR2-VCL led to PIP2 release from VCL, enabling PI3K activation via TLR2. These results highlight the complexity of TLR signaling, and the importance of discovering protein-protein interactions that contribute to the outcome of infection.

## Author summary

*Porphyromonas gingivalis* is an anaerobic bacterium strongly associated with periodontitis. Remarkably, *P. gingivalis* flourishes in the presence of inflammation and activated innate immune cells. *P. gingivalis* is sensed by TLR2, however, sensing leads to escape from macrophage bactericidal activity in a manner dependent on PI3K/Akt signaling. We used chemical cross-linking to identify induced interacting partners of TLR2 that contribute to bacterial immune escape, and found that infection induces TLR2 association with vinculin (VCL). Modeling the TLR2-VCL interaction highlighted critical amino acid residues of the TLR2 intracellular domain that were then experimentally validated by site-directed mutagenesis. We then confirmed that the induced TLR2-VCL interaction

relevant data are within the manuscript and its Supporting Information files.

**Funding:** This work was supported by a grant from the Israel Science Foundation (grant 1391/17 to GN). The funders had no role in study design, data collection and analysis, decision to publish, or preparation of the manuscript.

**Competing interests:** The authors have declared that no competing interests exist.

contributes to *P. gingivalis* immune evasion by enabling PI3K/Akt activation. Our results demonstrate the importance of induced TLR2 intracellular interacting partners in orchestrating downstream signaling that influences the outcome of the host-pathogen encounter.

## Introduction

Periodontitis is characterized by severe gingival inflammation leading to resorption of the tooth-supporting soft and hard tissue [1]. The oral microbiome in periodontitis is dysbiotic, with increased numbers of Gram-negative anaerobic organisms that thrive in an inflammatory environment [2]. *Porphyromonas gingivalis* is a Gram-negative, anaerobic, assacharolytic bacterium that disrupts the homeostasis of the subgingival region of the periodontium leading to dysbiosis [3]. The initiation and advancement of periodontal tissue destruction involves plaque formation, release of bacterial constituents, and most importantly, the host inflammatory response [4]. *P. gingivalis* invades epithelial and immune cells, where it can remain viable and replicate [5]. *P. gingivalis* has evolved to elude or subvert constituents of the host immune system [6,7]. Host Toll-Like Receptor 2 (TLR2) plays a major role in epithelial and immune cell recognition and response to live *P. gingivalis*, although additional TLRs and other innate receptors also contribute to the host response to *P. gingivalis* or its constituents [8–12] However, TLR2 signaling is manipulated by *P. gingivalis* to its advantage, as evidenced by the efficient clearance of the bacteria in vitro and in vivo when TLR2 is blocked or absent [8,13–15]. Furthermore, TLR2 knock-out mice are resistant to experimental periodontitis induced by *P. gingivalis* infection [8]. Canonical TLR2 signaling requires the intracellular adaptor protein Myeloid differentiation primary response (MyD88) to clear intracellular *P. gingivalis* [15,16]. To avoid bactericidal activity, *P. gingivalis* induces cross-talk between TLR2 and the complement receptor C5aR leading to a MyD88-independent signaling pathway that requires phosphoinositide-3-kinase (PI3K) [14,15]. In cells that possess MyD88, *P. gingivalis* induces MyD88 ubiquitination and proteasomal degradation, thereby avoiding bactericidal activity [15,17]. However, the protein-protein interactions downstream of TLR2 that determine whether non-canonical TLR2-PI3K signaling is initiated or not, are for the most part unknown.

Focal adhesions (FAs) connect cells to the extracellular matrix (ECM) via integrin receptors, and physically connect to the actin cytoskeleton through the recruitment of numerous FA-associated macromolecular proteins. Cytosolic components of FAs include VCL, talin, α-actin, paxillin, tensin, zyxin and Focal Adhesion Kinase (FAK). FAs exert profound effects on intracellular signaling by regulating signaling intermediates (e.g., Akt/PKB, MAP kinases and small GTPases) [18]. TLR2 signaling closely associates with integrin activation, and *P. gingivalis* interacts with TLR2/TLR1 and leads to inside-out signaling for the activation and binding of CD11b/CD18 in macrophages [19]. Here, we used a non-biased cross-linking and mass spectrometry approach and identified VCL as an interacting partner of TLR2 in the macrophage response to *P. gingivalis* infection. VCL is implicated in the cellular response to multiple infectious agents, but its role in the immune evasion of *P. gingivalis* remains unexplored. We elucidate structural aspects of TLR2-VCL interaction and its functional role in the response to bacterial challenge.

## Results

### TLR2 and VCL interact in response to bacterial challenge

To identify proteins interacting with TLR2 in response to *P. gingivalis*, THP1 macrophages were infected with bacteria at multiplicity of infection (MOI) 10 for 30 min and then protein interactions were stabilized using the membrane-permeable crosslinker dithiobis (succinimidyl propionate) (DSP) that contains a central thiol-cleavable bond. Native TLR2 was immunoprecipitated with anti-TLR2 bound to protein G beads from non-infected (control) and infected cells, and TLR2-binding partners were eluted by cleaving the cross-linker with dithiothreitol (DTT), leaving the TLR2 attached to the beads. The DTT eluate and the beads were analyzed by mass spectrometry (MS). As expected, TLR2 was identified on the beads of precipitates from both control and *P. gingivalis*-infected cells (6 and 7 unique peptides, respectively) and a TLR2 peptide that underwent mass shift due to half of the DSP cross-linker (carbamidomethyl (CAM) modification) was found in both cases (S1 Table). In the DTT eluates, TLR2 was either not present or represented by only 2 peptides suggesting that the majority of the protein remained bound to the beads. We next focused on Vinculin (VCL), a cytoskeleton protein essential for focal adhesion complex (FAC) formation, that was identified in both the bead fraction and DTT eluate of infected cells, and that contained the CAM modification (S1 Table). To validate the induced interaction between TLR2 and VCL, TLR2 was immunoprecipitated from control and *P. gingivalis*-stimulated macrophages after cross-linking with DSP, and the immunoprecipitates were analyzed for VCL by immunoblot (Fig 1A). In the absence of the DSP cross-linker VCL was not immunoprecipitated with TLR2 (S1A Fig). VCL also associates with TLR2 in primary human macrophages, however the difference between control and *P. gingivalis* stimulation is less pronounced (S1B Fig). Next, the colocalization of TLR2 and VCL in response to bacterial infection or to a TLR2 ligand Pam$_3$-Cys-Ser-(Lys)$_4$ (PAM) stimulation was visualized by co-immunofluorescence in THP1 macrophages (Fig 1B), and in TLR2-YFP transfected HeLa cells, widely used for VCL studies (Fig 1C). Colocalization was quantified by the JACoP toolbox under NIH ImageJ using Pearson and Manders' coefficient correlation [20]. This object-based fluorescent intensity analysis showed induced colocalization between TLR2 and VCL in response to TLR2 stimulation in both phagocytic (THP1) and non-phagocytic (HeLa) cells.

### Computational analysis of TLR2-VCL interaction

Our results suggested that TLR2 and VCL interact directly. We next used computational methodologies to study this interaction and to identify a region of TLR2 that binds VCL. Residues located at protein interfaces present distinct physico-chemical properties that can be identified without the presence of a hypothetical partner. We used Optimal Docking Area (ODA), a method for the prediction of residues involved in protein-protein interactions, that identifies regions (if any) with favorable energy change when buried in a protein-protein interface [21]. Fig 2A shows the results of ODA prediction mapped to the TLR2 TLR/IL-1 receptor (TIR) domain structure (PDB 1FYW) [22]. Notably, ODA identifies a highly probable protein-protein interface region including residues W684 and F719 and several of their neighbors. Then, we used both carry-on docking simulation with pyDock [23] between the 3D structures of TLR2 TIR domain and VCL (PDB 1TR2) [24]. pyDock computes the most probable (lowest energy) docking orientation between them. Although the docking results show different possible TIR domain orientations facing VCL with similar energy, many docking poses show W684 and F719 facing VCL (S2 Fig). Furthermore, from the 100 best PyDock conformations we computed NIPS (Normalized Interface Propensity) values, which correlate with the frequency

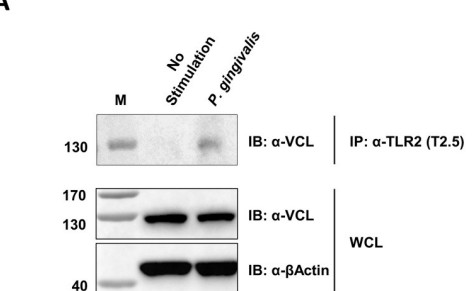

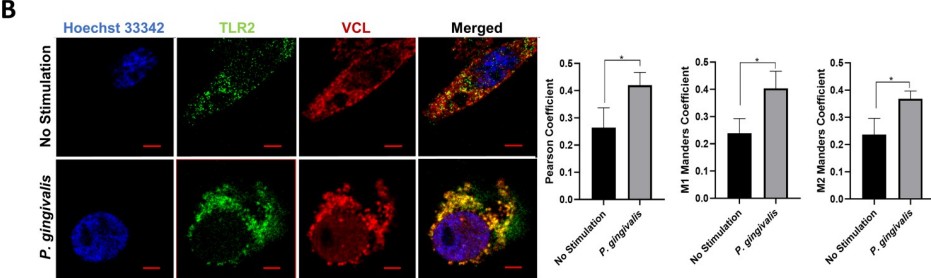

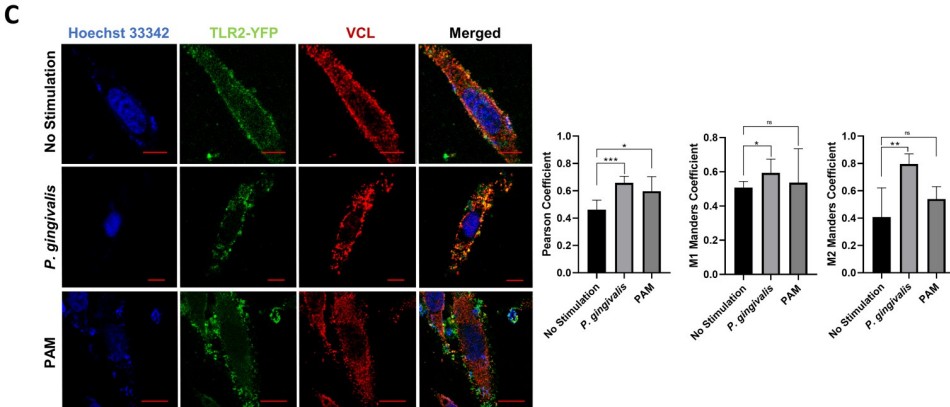

**Fig 1. TLR2 physically associates with VCL *in vitro*.** (**A**) Human macrophage THP1 cells were infected with *P. gingivalis* (MOI 10) for 30 min followed by cross-linking with DSP (M, marker). TLR2 was immunoprecipitated (IP) and eluates were analyzed for VCL by immunoblot (IB). Whole cell lysates (WCL) were analyzed to control for protein input and loading. (**B**) THP1 cells were differentiated for three days and left untreated or infected with *P. gingivalis* at MOI 10 for 30 min. Fixed cells were stained for endogenous TLR2 (green) and VCL (red), and nuclei were counterstained with Hoechst (blue). Data are representative of three independent experiments. (**C**) HeLa cells were transfected with TLR2-YFP (green). Control or treated cells (PAM vs. *P. gingivalis* at MOI 10 for 30 min) were stained for endogenous VCL (red), and nuclei were counterstained with Hoechst. Images were captured using a NIKON confocal microscope at 60X magnification. Yellow color indicates the co-localization of green and red channels. Co-localization was quantified using JACoP/ImageJ analysis software. Data are representative of three independent experiments. (ns: non-significant; *P $\leq$ 0.05; **P $\leq$ 0.01; ***P $\leq$ 0.001).

of a given residue to be located at the interface [25]. In order to find core residues of this interface (residues which contribute the most to the binding free energy) [26], we defined residues with ODA < -15 Kcal and NIP > 0.3 as core residues. In this way, we identified a cluster of five residues W684, I685, L717, F719 and S720 which are critical for the interaction of TLR2 with VCL (see Fig 2A and Table 1). These residues lie outside the TIR domain BB-loop region

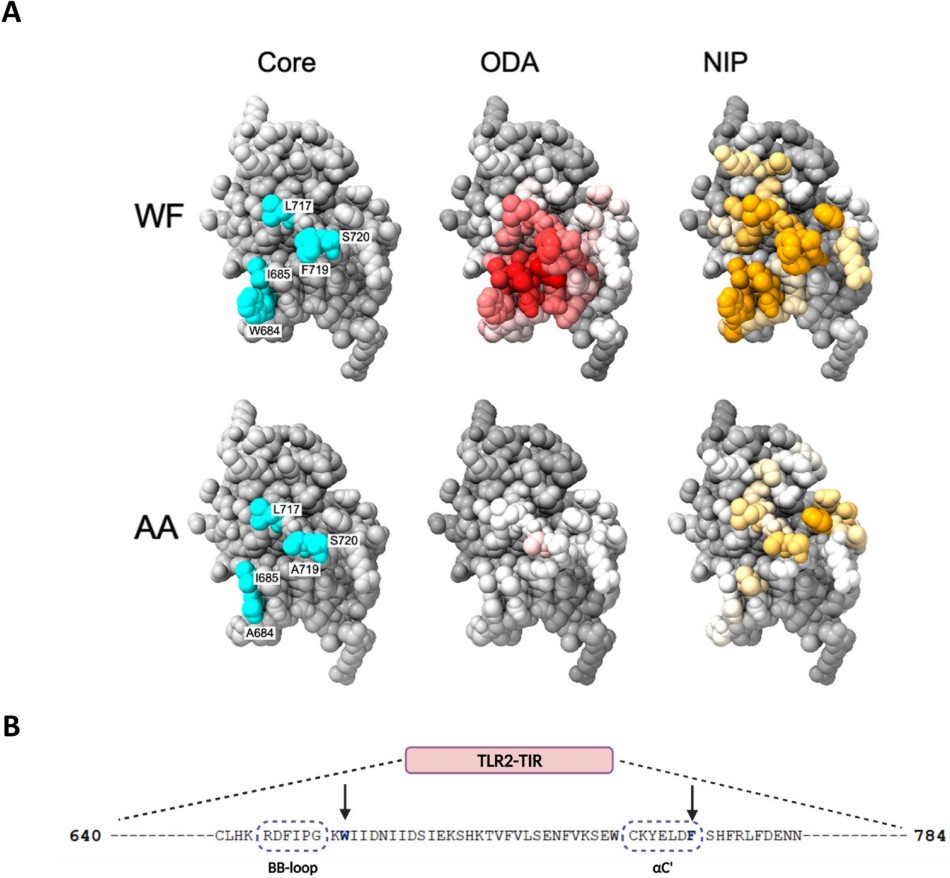

**Fig 2. Computational predictions of protein-protein interface residues.** (**A**) Five core residues of the TLR2 TIR domain are colored blue and labeled ("Core" column). The top row shows the structure for the native (W684, F719), and the second row for the alanine mutated (W684A, F719A), sequence. ODA and NIPs residue values colored on the TLR2 TIR domain crystal structure and core residue list. ODA values are colored in grey-white-red scale (decreasing values), where red indicates lowest energy values (residues with highest contribution to a protein-protein interface stability). NIPs values are colored with a grey-white-orange scale (increasing values), where higher NIP values represent a higher chance of a residue to participate in the specific TLR2-VCL interface. (**B**) TLR2-TIR domain sequence showing the two residues mutated to alanine by site-directed mutagenesis (arrows), and their proximity to TIR regions implicated in homo- and hetero-oligomerization (the BB loop and αC' helix). Fig 2B was created with BioRender.com.

**Table 1. Core residues of TLR2 TIR domain critical for TLR2-VCL interaction.**

| WT | NIPS | ODA | AA mutant | NIPS | ODA |
|---|---|---|---|---|---|
| W684 | 0.60 | -28.1 | A684 | 0.07 | -5.68 |
| I685 | 0.32 | -34.9 | I685 | 0.05 | -6.8 |
| L717 | 0.33 | -29.2 | L717 | 0.12 | -15.3 |
| F719 | 0.61 | -34.0 | A719 | 0.15 | -19.8 |
| S720 | 0.39 | -28.4 | S720 | 0.14 | -10.3 |

ODA and NIPS values for WT and W684A/F719A double mutant of core protein-protein residues of TLR2-VCL interface.

that is known to play a major role in the physical associations of TLRs with cytosolic adapter proteins (Fig 2B).

Next, W684 and F719 were mutated to Ala (A) individually or together (W684A, F719A, and the double mutant WF to AA) in a TLR2-YFP plasmid. I686, L717, and S720 were not targeted due to their proximity to W684 and F719. Since mutations can affect protein folding or localization, we tested the function of the constructs in an NF-κB -reporter cell line in response to the synthetic bacterial lipopeptide TLR2 ligand PAM. NF-κB activation of cells transfected with the native TLR2 plasmid vs. each of the mutated constructs was equivalent, demonstrating that the mutated TLR2 proteins fold and localize properly (Fig 3A). Overexpression of the TLR2-mutated proteins led to similar cellular localization (S3 Fig). HeLa cells were transfected with wild-type or mutant TLR2-YFP and then TLR2-VCL co-immunofluorescence was analyzed by staining for endogenous VCL. Naive and mutated transfected TLR2 proteins were expressed on the cell surface, and co-localization was measured using JACoP toolbox under NIH ImageJ [20]. When compared to the wild-type, F719A and the double mutant failed to co-localize with VCL, whereas W684A TLR2 co-localization was not significantly different from the native TLR2 (Fig 3B). To rationalize these experimental findings, we modelled the change in binding free energy (interface $\Delta\Delta G = \Delta G_{mutant}-\Delta G_{WT}$) induced by the AA double mutation using pyDock and FoldX. PyDock showed a very large change in energy of ~+12 Kcal/mol for the best three free docking results with the double AA mutant vs WF, indicating a clear destabilizing effect, and confirming that W684 and F719 are crucial to stabilize the interface (see Fig 2B and Table 1). Finally, FoldX computations show that the change to A of any of those residues has almost no impact on the stability of the TLR2 TIR domain fold (energies ~0.004 for the $\Delta\Delta G$ upon mutation), making it reasonable to assume the impact on the complex stability derives from TLR2-VCL impaired interaction and not from a folding disruption.

Taken together, our computational results suggest W684 and F719 are key players for the interaction of TLR2 intracellular domain with VCL. Their replacement considerably reduces NIPs and ODA values for W684 and F719, but also notably for all interface core residues (see Table 1), thus seriously impairing the ability of TLR2 to bind VCL, causing a partial or complete interface destabilization and functional disruption that we see experimentally.

## Split ubiquitin system validates induced TLR2-VCL interaction

To validate the inducible interaction of TLR2 with VCL in response to infection with *P. gingivalis*, we next adapted a split-ubiquitin two-hybrid system originally designed to study inducible interactions between membrane proteins (Mammalian Membrane Two-Hybrid system, MaMTH) [27,28]. In this system, TLR2 serves as the bait protein and is expressed fused with a C-terminal fragment of ubiquitin followed by the GAL4 transcription factor (TLR2-Cub-TF). Prey proteins are expressed fused to N-terminal ubiquitin, and upon interaction of bait and prey proteins, pseudoubiquitin is formed and cleaved by cellular deubiquitinases, releasing the TF. Bait and prey plasmids are overexpressed in cells that express luciferase under the GAL4 promoter (Fig 4A). To confirm that the MaMTH system can be used to study inducible interactions between TLRs and their cytosolic partners, we cloned TIRAP and IRAK-4 into the bait vector as positive and negative controls, respectively. TIRAP is an adaptor protein that interacts directly with TLR2 through TIR-TIR interactions and recruits MyD88 to initiate Myddosome formation [29]. IRAK-4 binds to the Myddosome multiprotein complex and is critical for TLR2 signaling, however it does not interact directly with TLR2 [30]. Bait and prey plasmids contain additional tags to confirm expression following transfection (S4A and Fig 4B). HEK293T cells stably expressing luciferase following five *GAL4* upstream activating sequence

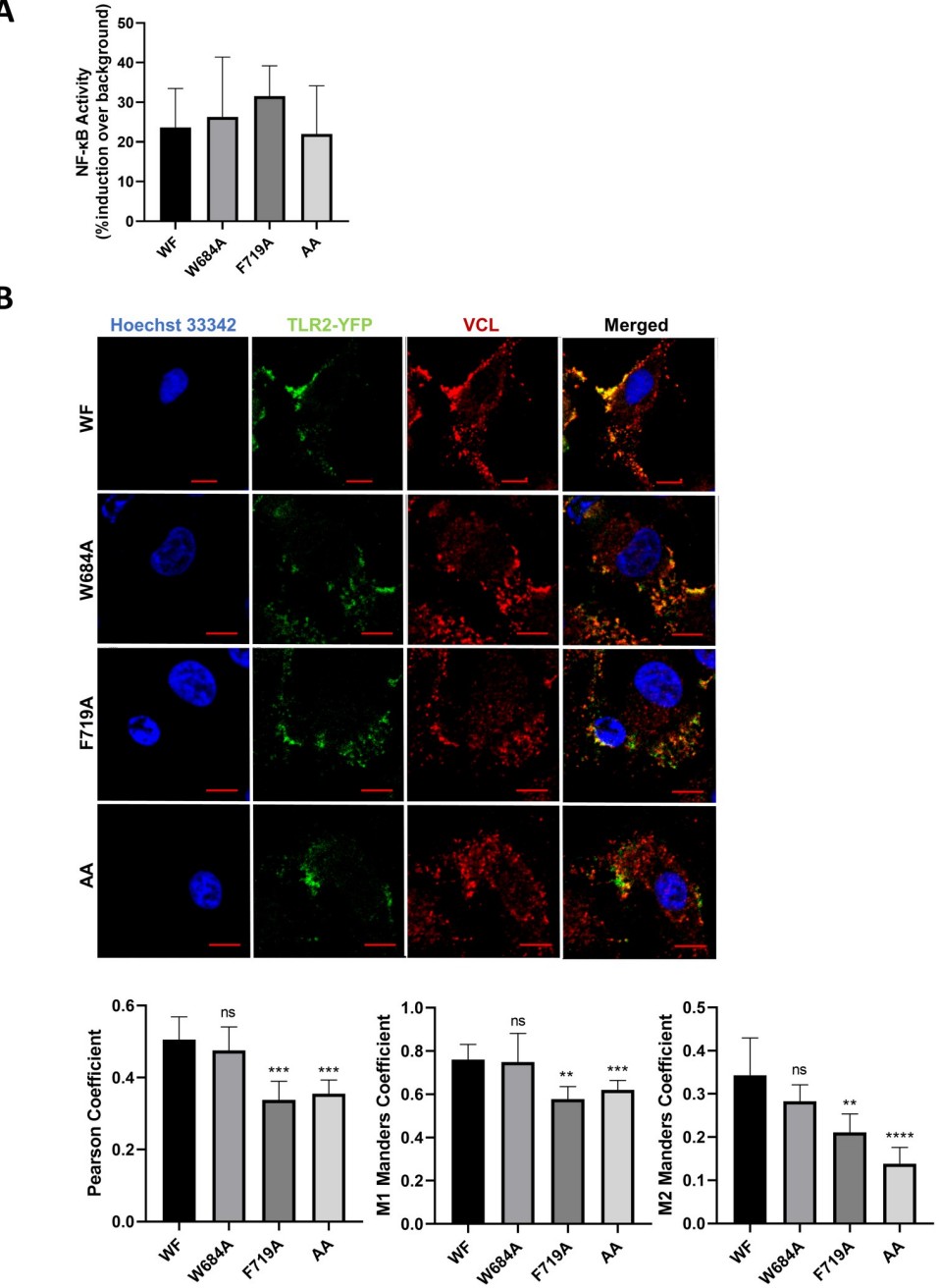

**Fig 3. Site directed mutagenesis validates TLR2 residues involved in TLR2-VCL interaction. (A)** NF-κB reporter-HEK293 cells were transfected with native TLR2 (W684/F719), or each of the TLR2 mutants W684A, F719A, or W684A/F719A together with TLR1 and MyD88, and then cells were stimulated with PAM (100ng/mL) for 4 h. Cells were lysed and NF-kB activation was measured by luciferase activity. Percent NF-kB induction is shown relative to luciferase values of control cells transfected in an identical manner but not activated with PAM. Data represent mean values of three independent experiments. **(B)** HeLa cells were transfected with YFP-fused native (WF) TLR2, or W684A, F719A, or AA TLR2 mutants (green). Cells were fixed and stained for VCL (red) and nuclei were stained with Hoechst (blue). Images were captured using a NIKON confocal microscope at 60X magnification. TLR2-VCL co-localization was analyzed using JACoP/ImageJ analysis software. Data are representative of five independent experiments. (ns: non-significant; **P ≤ 0.01; ***P ≤ 0.001; ****P ≤ 0.0001).

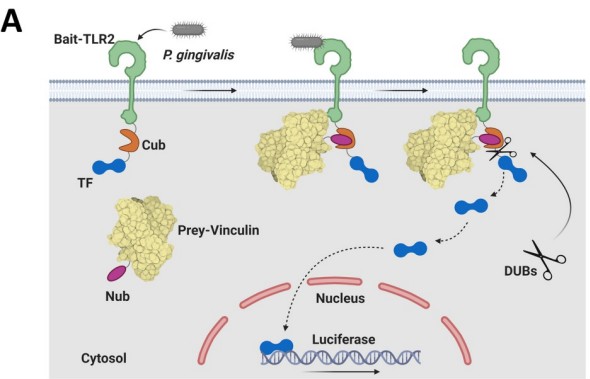

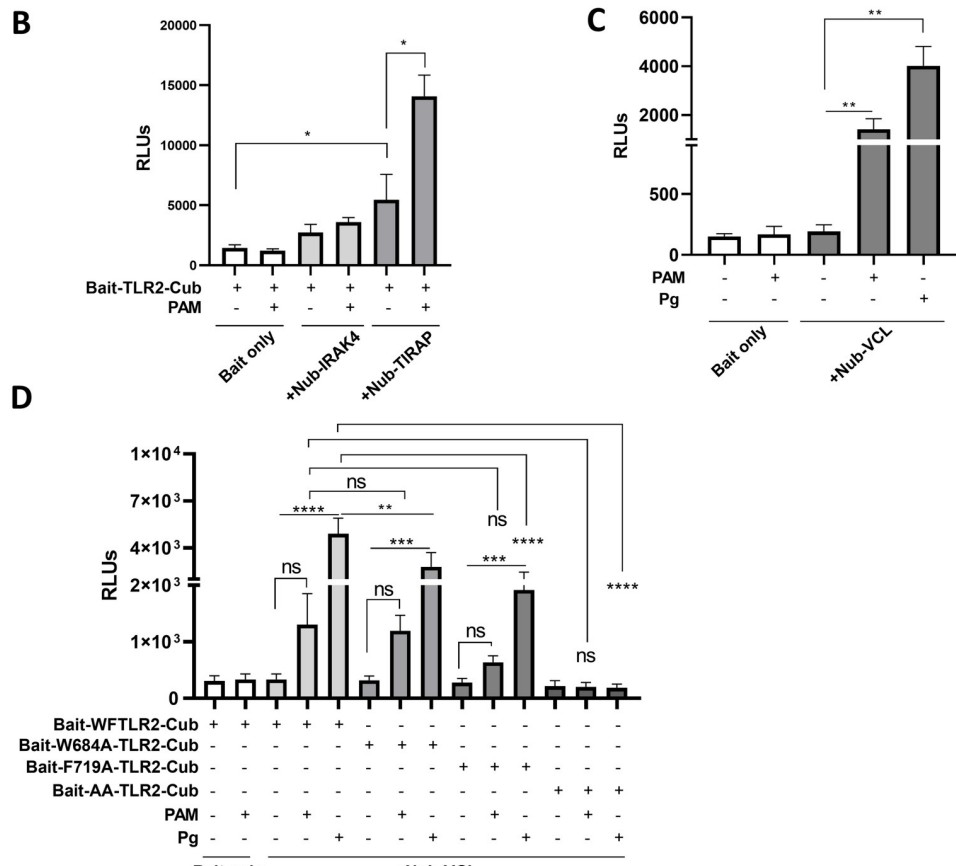

**Fig 4. MaMTH system for validating TLR2-VCL interaction.** (**A**) Graphical summary of MaMTH split ubiquitin system for detection of bacterially-induced TLR2-VCL interaction (**B**) Luciferase activity of untreated or PAM-stimulated HEK293T cells co-transfected with TLR2-bait and the indicated positive and negative control prey constructs. Data are representative of five independent experiments. (**C**) HEK293T cells co-transfected with TLR2-bait and VCL prey plasmids were left untreated, exposed to PAM (100ng/mL), or infected with *P. gingivalis* (MOI 10) for 8 h. Data are representative of four independent experiments. (**D**) Luciferase activity in HEK293T cells co-transfected with wild-type TLR2 bait, or mutants TLR2 bait mutants W684A, F719A or WF, together with VCL prey. Transfected cells were untreated, treated with PAM (100ng/mL), or infected with *P. gingivalis* (MOI 10) for 8 h. Data are representative of two independent experiments. (ns: non-significant; *P≤0.05; **P ≤ 0.01; ***P ≤ 0.001; ****P ≤ 0.0001). Fig 4A was created with BioRender.com.

repeats (5×GAL4UAS-luciferase) were co-transfected with TLR2-Cub-TF and either TIRAP-Nub (positive control) or IRAK4-Nub (negative control). Transfection with TIRAP-Nub, but not IRAK4-Nub, induced luciferase activity significantly, reflecting the known protein-protein interaction of TLR2 and TIRAP. PAM stimulation of transfected cells induced a further significant increase in the luciferase signal in the positive control TIRAP-Nub transfected cells, but not the negative control IRAK4-Nub transfected cells (Fig 4B). We next transfected reporter cells with TLR2-Cub-TF and VCL-Nub and stimulated the cells with PAM or *P. gingivalis*. Bacterial infection, or stimulation with PAM, induced a significant signal in the reporter cells, confirming the inducible interaction between TLR2 and VCL (Fig 4C). We then utilized the assay to reconstruct the interaction of TLR2 mutants with VCL by creating different site-directed mutations on bait-TLR2-Cub (W684A, F719A and the double mutant-AA). All the mutants were equally expressed and localized to the membrane (S4B Fig). Consistent with the co-immunofluorescence result, the F719A mutation significantly reduced the induced interaction of TLR2 with VCL, whereas the W684A mutation did not. However, the contribution of W684 is observed in combination with F719 since the double mutation of W684A and F719A (AA) abolished the inducible luciferase signal completely (Fig 4D). Taken together, our results show that bacterial infection, or PAM stimulation, induces TLR2 interaction with VCL through specific residues of the TIR domain outside the BB loop.

## Macrophage TLR2- and TLR4-dependent cytokine production is dampened by VCL

In macrophages, TLR2-mediated sensing of *P. gingivalis* results in cytokine production [8]. We next examined the effect of VCL knock-down on the macrophage cytokine response to infection with *P. gingivalis in vitro*, and compared this response to that induced by the proto-typical TLR2 ligand PAM, which stimulates TLR2 in a strictly MyD88-dependent manner, and to a TLR4 ligand, LPS (lipopolysaccharide). VCL knock-down was achieved by lentiviral shRNA in THP1 cells (shVCL), and reduced VCL expression was confirmed by western blot and quantitative real-time PCR (RT-PCR) (Fig 5A). Tumor necrosis factor (TNF) production by shVCL cells was enhanced compared to control shRNA cells (shCtrl) in response to all of the stimulants (Fig 5B), indicating that VCL reduces TLR2 and TLR4 signaling that leads to TNF production. RT-PCR confirmed the inhibitory effect of VCL on cytokine gene regulation in response to TLR2 or TLR4 stimulation (Fig 5C).

## VCL is involved in the uptake and survival of *P. gingivalis* in macrophages

We next tested the role of the induced TLR2-VCL interaction in the phagocytosis and intracellular fate of *P. gingivalis*. We compared the intracellular survival of *P. gingivalis* in differentiated shVCL and shCtrl THP1 macrophages. Following infection, extracellular bacteria were killed by antibiotic treatment, and macrophages were incubated for an additional hour, lysed, and surviving intracellular bacteria were enumerated by plating lysates on blood agar. Interestingly, VCL knock-down significantly increased *P. gingivalis* intracellular survival (Fig 5D). Intracellular survival in shVCL macrophages was completely abrogated by blocking TLR2, confirming that VCL suppresses the TLR2-dependent bacterial survival pathway (Fig 5D). We next asked whether increased intracellular survival could be explained by enhanced phagocytosis/*P. gingivalis* invasion in the absence of VCL. Differentiated shVCL THP1 and shCtrl THP1 cells were infected with fluorescein isothiocyanate (FITC)-labelled *P. gingivalis* and then cells were fixed and analyzed by flow cytometry. In contrast to our expectation, VCL knock-down strongly reduced the percentage of cells that phagocytosed *P. gingivalis* as well as the overall fluorescence intensity of the population, reflecting the number of bacteria phagocytosed per

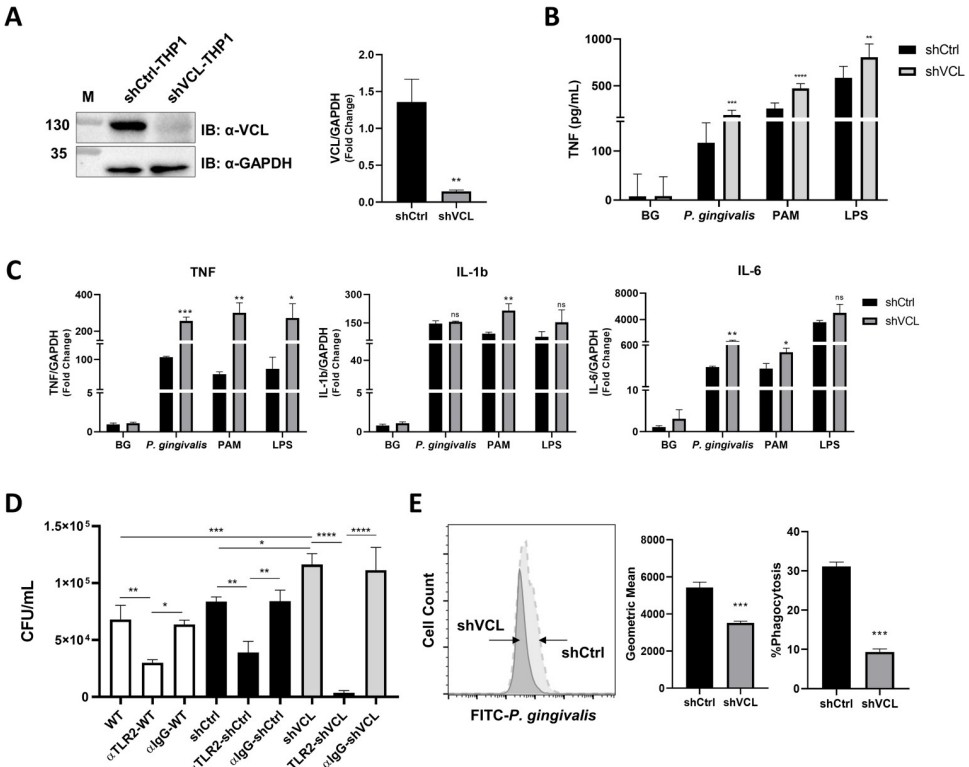

**Fig 5. Cytokine production and bacterial survival in VCL knock-down macrophages. (A)** Representative IB of VCL and GAPDH in whole cell lysates from shCtrl and shVCL-THP1 cells, and RT-PCR showing levels of VCL expression **(B)** TNF production by shCtrl and shVCL-THP1 cells challenged with *P. gingivalis* (MOI 10), PAM (10ng/mL), and LPS (10ng/mL) for 4 h, was measured by ELISA. Data are representative of five independent experiments. Unstimulated cells were used as controls (BG, background). **(C)** PMA-differentiated shCtrl and shVCL-THP1 cells were challenged with *P. gingivalis* (MOI 10), PAM (10ng/mL), and LPS (10ng/mL) for 4 h and mRNA levels of TNF, IL6, and IL1b were quantified by quantitative RT-PCR. mRNA from unstimulated cells collected at the same time point was used as background (BG). Data are representative of three independent experiments. **(D)** Intracellular *P. gingivalis* survival was determined in wild-type (WT) THP1, shCtrl-THP1 and shVCL-THP1 cells using the antibiotic protection assay as per methods with and without prior blocking of TLR2 using T2.5 anti-TLR2 mAb (αTLR2) vs. IgG1 isotype control. CFU were enumerated after 7 days of anaerobic growth (ND, none detected). Data are representative of three independent experiments. **(E)** shCtrl and shVCL-THP1 cell phagocytosis of FITC-labelled *P. gingivalis* was measured by flow cytometry (percentage of fluorescent cells above background, and geometric mean of fluorescence). A sample histogram is shown on the left. Data representative of three independent experiments are shown on the right. (ns: non-significant; *P≤0.05; **P ≤ 0.01; ***P ≤ 0.001; ****P ≤ 0.0001).

cell (Fig 5E). Therefore, VCL is required for efficient phagocytosis of *P. gingivalis*. Intracellular *P. gingivalis* survival is significantly higher in shVCL cells (Fig 5D) despite the reduced phagocytosis, suggesting that VCL strongly blocks *P. gingivalis* evasion from macrophage killing.

## VCL negatively regulates TLR2-dependent PI3K activation

*P. gingivalis* blocks phagolysosomal maturation leading to improved intracellular survival in macrophages by inducing TLR2 signaling through PI3K [14,15]. Therefore, we next determined the role of VCL in the activation of Akt phosphorylation downstream of PI3K activation, in response to *P. gingivalis*. We infected shVCL and shCtrl THP1 macrophages with *P. gingivalis* and lysed cells at various time points after infection. At each time point following infection, *P. gingivalis* induced greater Akt phosphorylation in shVCL-THP1 cells compared to shCtrl-THP1 cells (Fig 6A). To confirm that PI3K activation is driving *P. gingivalis*

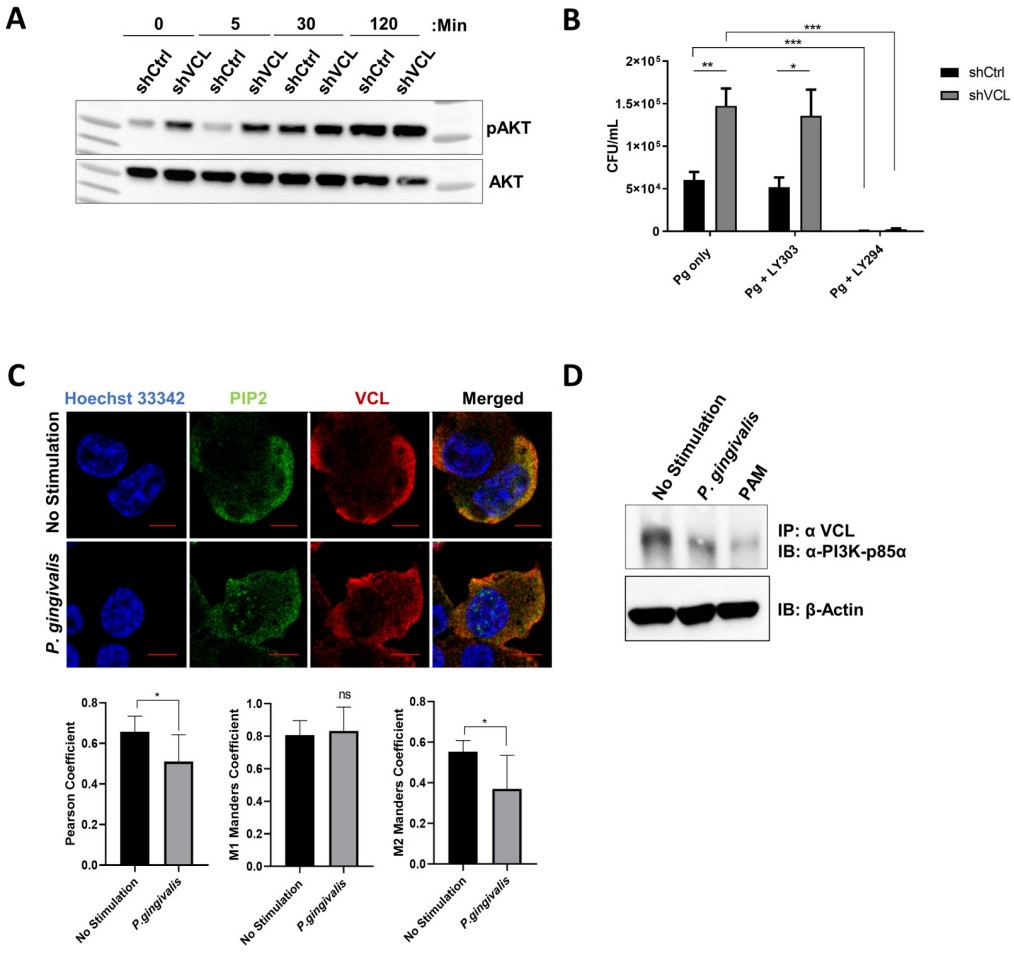

**Fig 6. VCL negatively regulates TLR2-dependent PI3K activation. (A)** shVCL and shCtrl THP1 cells were challenged with *P. gingivalis* (MOI 10) at different time points and lysates were analyzed by IB for phospho-Akt (P-Akt S473) and total Akt. Representative immunoblot of three independent experiments. **(B)** shCtrl-THP1 and shVCL-THP1 cells were pre-treated with the PI3K inhibitor LY294 or the control inhibitor LY303 prior to infection with *P. gingivalis* (MOI 10). Intracellular *P. gingivalis* survival was determined using the antibiotic protection assay. Data are representative of three independent experiments. **(C)** *P. gingivalis*-infected and control THP1 cells were fixed and stained for PIP2 (green), and VCL (red), and nuclei were stained with Hoechst (blue). Images were captured using a NIKON confocal microscope at 60X magnification. PIP2-VCL co-localization was analyzed using JACoP/ImageJ analysis software. Data are representative of two independent experiments. **(D)** Differentiated THP1 cells were infected with *P. gingivalis* for 30 min and VCL was immunoprecipitated. Eluates were analyzed by IB with an antibody to PI3k-p85α. Data are representative of three independent experiments. (ns: non-significant; *P≤0.05; **P ≤ 0.01; ***P ≤ 0.001).

intracellular survival in shVCL macrophages, similar to WT macrophages, we blocked PI3K using LY294 (vs. a control inhibitor, LY303), and followed *P. gingivalis* intracellular survival. Blocking PI3K in shVCL cells significantly decreased *P. gingivalis* survival (Fig 6B), correlating with the increased PI3K-Akt activation observed in these cells. Importantly, Akt phosphorylation was enhanced in shVCL cells at baseline, prior to infection with *P. gingivalis*, suggesting that VCL regulates PI3K in the steady-state. Several studies showed that the PI3K substrate phosphoinositol-2-phosphate (PIP2) mediates phagocytosis [31], and that VCL contains PIP2 binding sites [32]. We next hypothesized that VCL sequesters PIP2 making it unavailable for PI3K. Indeed, we found that VCL is co-localized with PIP2 in resting WT THP1 macrophages, and dissociates from PIP2 in response to *P. gingivalis* (Fig 6C). Since VCL has also been shown

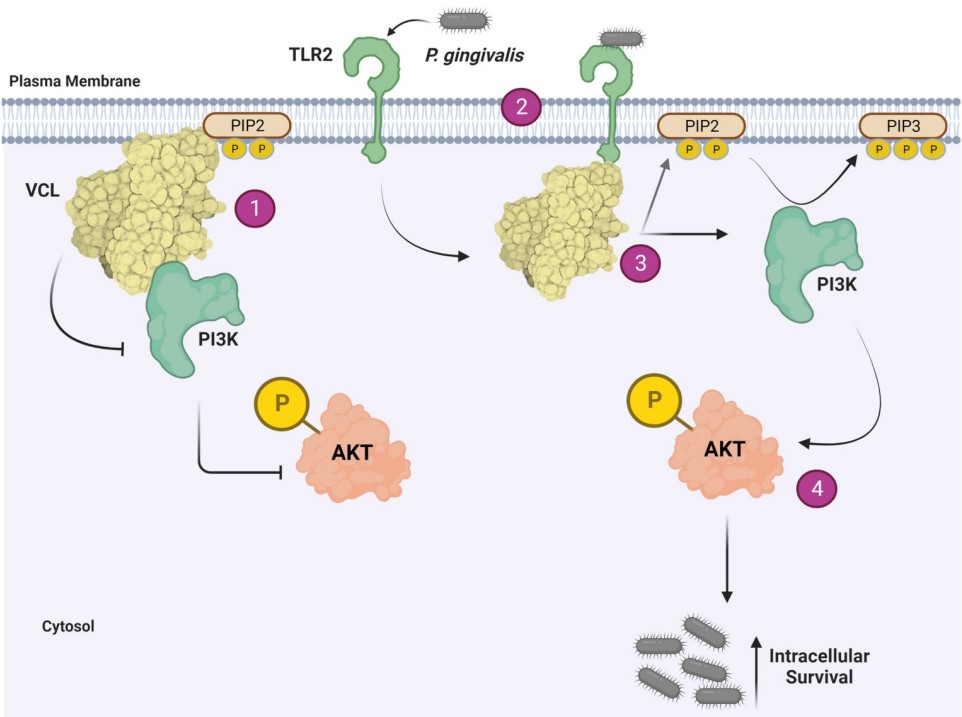

**Fig 7. Schematic illustration.** In un-infected macrophages [1], VCL associates with PIP2 and p85α, preventing activation of downstream Akt phosphorylation and thereby favoring macrophage killing of *P. gingivalis* when cells are infected. TLR2 senses *P. gingivalis* and is induced to interact with VCL [2], which causes VCL to dissociate from PIP2 and p85α [3]. This enables PI3K conversion of PIP2 to PIP3, and subsequent phosphorylation of Akt [4], leading to increased intracellular survival. Fig 7 was created with BioRender.com.

to bind class 1 PI3K [33], we next evaluated VCL and PI3K p85α association in THP1 macrophages. At steady state we found that p85α and VCL are associated in WT THP1 macrophages, and that *P. gingivalis* infection causes the complex to dissociate (Fig 6D). This suggests that in addition to reducing PIP2 availability, VCL negatively regulates TLR2-PI3K activation by directly binding PI3K p85α itself. The combined effect reduces TLR2-PI3K signaling, an effect that is counteracted by the induction of TLR2-VCL association following infection with *P. gingivalis* (Fig 7).

## Discussion

*P. gingivalis* is a human pathogen that thrives in the inflamed environment of the gingival crevice where it is in continuous communication with innate immune cells. Macrophages sense *P. gingivalis* through TLR2, however *P. gingivalis* manipulates the interactions downstream of TLR2, favoring activation of PI3K and escape from killing [15] [13,14]. We used a cell-penetrating cross-linker followed by immunoprecipitation of TLR2 and LC-MS to identify novel interacting partners of native TLR2 involved in the host response to *P. gingivalis*. Limitations to the approach can explain why we did not recover peptides derived from the known TLR2 membrane interacting partners TLR1 or TLR6, or classical TLR2 adapter proteins such as MyD88. Importantly, we only examined one time point of activation, and the interactome will likely shift over time with some time-points containing more of the known TLR2 interacting proteins. A complete map of the interactome would require stimulation at different time points and/or using a single time point with various MOIs. Membrane proteins also require

significant optimizations for detection by MS, due to their hydrophobic characteristics and lipid-rich environment [34]. In addition, the amino acid compositions of some proteins obstruct their discovery following cross-linking, since the cross-linking reactions alter the efficiency of enzymatic digestion [35]. Finally, although we were able to identify VCL association with TLR2 in primary human macrophages, TLR2 immunoprecipitation in these cells is less optimal and our studies rely mostly on THP1 macrophages.

Our findings reveal a role for the cytoskeletal protein VCL in the outcome of the crosstalk between *P. gingivalis* and human macrophages. Despite decreased phagocytosis in VCL knock-down macrophages, significantly more live *P. gingivalis* are recovered from inside cells when VCL is reduced (Fig 5D), attesting to the powerful role of VCL in thwarting the immune evasion strategy of *P. gingivalis*. Mechanistically, VCL suppresses PI3K activation by sequestering p85α and its substrate PIP2. The induced interaction between TLR2 and VCL diminishes these associations, thereby enabling TLR2-PI3K signaling. Since lipopeptide activation of TLR2 induced a similar (although weaker) interaction with VCL (Fig 4), this pathway may be more broadly involved in the macrophage response to pathogens recognized by TLR2. Interestingly, VCL is involved in the cellular response to multiple pathogens. VCL is necessary for *C. trachomatis* invasion [36], and contributes to the intracellular spread of *S. flexneri* by concentrating around bacterial clusters in the cytosol [37]. On other hand, VCL is dispensable for integrin-mediated uptake of *S. aureus* [38], and due to its negative effect on paxillin phosphorylation, VCL blocks the early stages of HIV-1 infection [38,39]. Therefore, VCL is an important regulator of the outcome of innate immune sensing of pathogens.

The role of VCL in TLR signaling can be understood in the broader context of interactions between TLR signaling pathways and the cytoskeleton. VCL is a member of the FAC whose members are associated with TLR signaling pathways. FAK interacts with the TLR adapter proteins MyD88 and TIRAP, and is important in recruitment of MyD88 for LPS-induced cytokine secretion [40]. The cell adhesion-activated tyrosine kinase PYK2 also associates with MyD88 and tightly regulates MyD88-dependent NF-κB activation and IL-1β expression induced by LPS in macrophages [41]. *P. gingivalis* itself was shown to hydrolyse actin by its lysine protease leading to apoptosis in gingival epithelial cells (GECs) [42], and FAK and paxillin are also targeted by *P. gingivalis* proteases, causing impaired wound healing and periodontal tissue regeneration [43]. Recent studies showed that talin1 is important for the preassembled TLR complex in dendritic cells (DCs) at steady state, and that talin1 regulates MyD88-dependent signaling via direct interaction with MyD88 and Phosphatidylinositol 4-Phosphate-5 kinase (PI5K) [44]. The absence of talin1 reduces TLR-stimulated cytokine production, in contrast to the absence of VCL that leads to enhanced cytokine production in response to both TLR2 and TLR4 ligands (Fig 5B). The availability of membrane PIP2, necessary for recruitment of TIRAP to TLR2 and TLR4, may explain this difference since talin1 facilitates production of PIP2 [44], whereas our data suggest that VCL may reduce PIP2 availability. In fact, VCL shifts from a closed form to an active open conformation upon monocyte to macrophage differentiation, and the open form binds PIP2 and other FAC components [45]. This conformational shift can thereby reduce TIRAP recruitment to the TLR signaling complex and subsequent cytokine production.

We adapted a split-ubiquitin system designed for the study of inducible membrane protein-protein interactions [27,28], to study TLR2 interactions with cytosolic proteins. Using known interacting proteins of TLR2 signaling (TIRAP and IRAK4) as positive and negative controls, we demonstrated that this system can be used to validate novel TLR interacting partners such as VCL. The physical associations of TLRs with cytosolic adapter proteins have focused on the BB loop of the TIR domain that is present in all TLRs and TLR adapter proteins [46,47]. The BB-loop joins the βB and αB strands and is a critical region of interaction of the dimeric

TIR-TIR interface for TLR interactions with adapter proteins [48]. We discovered and validated that VCL interacts directly with TLR2, however overlapping non-protein specific and VCL-specific computational approaches indicated a region outside the BB loop as the core region of the TIR domain interface with VCL. This approach highlights the potential protein-protein interactions directed by the TIR domain with proteins such as VCL that lack TIR domains. Importantly, site-directed mutagenesis studies confirmed the importance of the region defined computationally. In particular, double mutation of W684 and F719 abolishes any interaction of TLR2 with VCL observable by co-immunofluorescence or in the quantifiable split ubiquitin system. Single mutation studies indicate that F719 of the αC-helix is a more critical residue than W684 for the TLR2-VCL interaction. This region in TLR2 or other TLRs may be important for additional protein-protein interactions that impact on the host response to pathogens.

Periodontal disease is characterized by a dysbiotic microbiome and inflammation, ultimately leading to osteoclast activation and resorption of the alveolar bone that supports the teeth. We revealed a role for VCL in modulating the macrophage inflammatory response and improving bactericidal activity, effects that should be protective in periodontal disease in vivo. However, VCL is also required for osteoclast function, as evidenced by enhanced bone volume in osteoclast-lineage VCL knock-out mice [49]. Therefore, greater inflammation and bacterial survival in the setting of VCL reduction would be counterbalanced by inhibition of osteoclast-mediated bone resorption. Our findings highlight the different ways that the cytoskeletal protein VCL can influence a complex and common disease.

## Materials and methods

### Ethics statement

All blood samples were collected from healthy volunteers after informed consent (Hadassah Medical Organization Helsinki committee, approval no. HMO-078520).

### Reagents

LY294002 (reversible inhibitor of PI3K), LY303511 (inactive analog of LY294002), glutaraldehyde (GTA), digitonin, and *E. coli* lipopolysaccharide (LPS) were obtained from Sigma-Aldrich (Rehovot, Israel). Paraformaldehyde (PFA) was from Biolabs (Jerusalem, Israel). Pam$_3$-Cys-Ser-(Lys)$_4$ (PAM) was obtained from InvivoGen (CA, USA). ReadyTag anti-DDDDK was from BioXcell (PA, USA). V5 Tag Monoclonal Antibody (TCM5) was from eBioscience (CA, USA). GAPDH and β-Actin antibodies were from Cell Signaling Technology (MA, USA).

### Plasmids

pcDNA3-TLR2-YFP was a gift from Doug Golenbock (Addgene plasmid # 13016; http://n2t.net/addgene:13016; RRID: Addgene_13016). pcDNA3-TLR1-YFP was a gift from Doug Golenbock (Addgene plasmid # 13014; http://n2t.net/addgene:13014; RRID: Addgene_13014). pCMV-HA-MyD88 was a gift from Bruce Beutler (Addgene plasmid # 12287; http://n2t.net/addgene:12287; RRID: Addgene_12287) [50]. pDONR 221 Entry vector compatible for gateway cloning system was obtained from Dr. Maggie Levy (The Robert H. Smith Faculty of Agriculture, Food & Environment, The Hebrew University of Jerusalem). Bait and Prey destination gateway cloning vectors were a kind gift from Dr. Igor Stagljar (Biochemistry, University of Toronto) [27,28]. pGreenFire1-NF-κB (EF1α-puro) lentivector was from System Biosciences (CA, USA).

## Cell lines and bacterial culture

Human macrophage THP1 cells and HeLa cells were obtained from the American Type Culture Collection (ATCC, VA, USA). THP1 cells were maintained in RPMI (Sigma-Aldrich) supplemented with 10% fetal calf serum (FCS), 2 mM L-glutamine, penicillin (100 units/ml), streptomycin (100 µg/ml), 10 mM HEPES, and 1 mM sodium pyruvate (Biological Industries, Israel). Short Hairpin RNA (shRNA) mediated vinculin (VCL) knocked-down and shRNA control THP1 cells were maintained in the presence of 2 µg/ml puromycin along with other supplements. THP1 cells were differentiated with 5 ng/mL phorbol-12-myristate-13-acetate (PMA) (Sigma-Aldrich) for 72 h to obtain mature macrophages. HEK293 cells (ATCC) were maintained in Dulbecco's modified Eagle's medium (DMEM) (Sigma-Aldrich) supplemented with 10% fetal calf serum, 2 mM L-glutamine, penicillin (100 units/ml), streptomycin (100 µg/ml) (Biological Industries) (complete DMEM). Mammalian Membrane Two-Hybrid system (MaMTH) modified reporter HEK293T cell line stably expressing luciferase following five *GAL4* upstream activating sequence repeats (5×GAL4UAS-luciferase) was a gift from from Dr. Igor Stagljar (Biochemistry, University of Toronto) [27,28] and maintained in complete DMEM. The cell lines were cultured at 37°C, 100% humidity, and 5% $CO_2$ atmosphere. *P. gingivalis* (strain 381) was cultured in anaerobic conditions at 37°C in Wilkins Chalgren Anaerobe Broth (Oxoid Ltd, Hampshire, UK). For infection of cell lines, bacteria were collected by centrifugation, washed with sterile PBS, and colony forming unit (CFU) were determined by optical density (OD) at 650 nm (OD of 0.1 was determined to be $10^{10}$ CFU/mL) [8]. Bacteria were diluted in cell culture medium and added to cells at the indicated multiplicity of infection (MOI).

## Isolation of PBMCs

Human peripheral blood mononuclear cells (PBMCs) were isolated from fresh whole blood by Ficoll gradient centrifugation, according to the standard density gradient centrifugation methods following manufacturer's protocol (Ficoll-Paque, Amersham Biosciences, UK). Isolated PBMCs were differentiated into mature macrophages using macrophage colony-stimulating factor (M-CSF, 20 ng/mL, Peprotech, Israel) for 6 days in RPMI.

## Generation of lentiviral cell lines

pLKO.1 harboring shRNA specific to human VCL and control shRNA were a generous gift from Professor Christof R. Hauck (Universität Konstanz, 78457, Konstanz, Germany). HEK293T cells were co-transfected with corresponding pLKO.1 plasmid or pGreenFire1-NF-κB lentivector (co-expressing GFP and luciferase in response to NF-κB activity) along with plasmids encoding 3rd generation lentiviral components. Viral-particles were collected at 36 h and 48 h after transfection, filtered through 0.45 µm filter, and stored at −80°C. THP1 cells were transduced with shRNA containing lentiviral particles and selected with 2 µg/mL puromycin (Sigma-Aldrich). HEK293 cells were transduced with NF-κB reporter lentiviral particles and selected with 2 µg/mL puromycin (Sigma-Aldrich).

## Immunoprecipitation and analysis

THP1 macrophages, or primary PBMC-derived macrophages, were infected with *P. gingivalis* for 30 min and then proteins were cross-linked in the live cells with DSP (dithiobis(succinimidyl propionate)) (ThermoFisher scientific, MA, USA) at 4°C for 2 h [51]. Cross-linking was terminated by incubation with 20 mM Tris, pH 7.5, for 15 min. Cell lysates were immunoprecipitated with mouse anti-TLR2 antibody (clone T2.5) (Hycult Biotech, Netherland) or anti-

VCL (Abcam, Cambridge, UK) bound to protein G beads (ThermoFisher) and eluted either with DTT for mass spectrometry (MS) analysis or with glycine pH 2.5 for immunoblot (IB) analysis. For MS, eluted proteins and proteins bound to the beads were digested with trypsin and analyzed by Liquid Chromatography (LC)-MS/MS on Q-Exactive (ThermoFisher) and then identified by Discoverer 1.4 software with the search algorithm Sequest against the human proteome from the Uniprot database, and a decoy database (in order to determine the false discovery rate). All the identified peptides were filtered with high confidence (above 1% false discovery rate threshold), top rank, and mass accuracy. Common epithelial cell contaminants were removed. Proteins identified by single peptides were not considered as certain. MS analysis was performed at the Smoler Proteomics Center, Technion Israel Institute of Technology (Haifa, Israel). For IB, immunoprecipitates were analyzed for TLR2 (polyclonal anti-TLR2, ThermoFisher), VCL (Abcam) or PI3 Kinase p85α (Cell Signaling Technology) followed by horseradish peroxidase (HRP)-conjugated polyclonal Goat anti-rabbit IgG (Abcam).

### Cytokine analysis

Cytokine levels were determined by enzyme-linked immunosorbent assay (ELISA) using human Elisa MAX sets (Biolegend, CA, USA) for tumor necrosis factor (TNF), according to the manufacturer's instructions.

### RNA extraction and real-time PCR

Total RNA was extracted from cells using TRIzol Reagent (Sigma-Aldrich). 1 μg RNA sample was reverse-transcribed into cDNA using qPCRBIO cDNA synthesis kit (PCRBIOSYSTEMS, PA, USA). The PCR primer sequences are listed in S2 Table. Quantitative PCR was performed using SYBR green mix (Bio-Rad, CA, USA), and expression values were normalized to GAPDH and calculated based on the Cq ($^{\Delta\Delta Cq}$) method [52]

### Akt Phosphorylation Assay

$1x10^6$ shVCL and shCtrl THP1 cells/well were plated in a 6 well plate and differentiated with PMA for 72 h followed by overnight with 1% FCS. Cells were exposed to *P. gingivalis* for different time periods and then cell lysates were prepared in radioimmunoprecipitation assay (RIPA) buffer containing protease and phosphatase inhibitors (Sigma-Aldrich). Total and phosphorylated levels of Akt were measured by IB (antibodies from Cell signaling Technology).

### Intracellular survival assay

$1x10^6$ shCtrl and shVCL THP1 cells/well were differentiated with PMA for 72 h followed by challenge with *P. gingivalis* (MOI 10) for 1 h. Cells were treated with LY294002, LY303511 or anti-TLR2 (clone T2.5) vs. IgG1 isotype control (Biolegend) for 1 h prior to challenge with *P. gingivalis*. Extracellular bacteria were killed by treatment with 6 mg/mL Metronidazole (Sigma-Aldrich) and 0.3 mg/mL Gentamycin (Biological Industries) for 1 h. The medium was then replaced with complete DMEM, cells were incubated for 1 h and then lysed in sterile distilled water for 20 min. Lysates were plated on blood agar (Novamed, Jerusalem, Israel) in anaerobic conditions for 7 to 10 days and CFU were enumerated.

### Phagocytosis assay

*P. gingivalis* was labeled with fluorescein isothiocyanate (FITC) by incubating washed bacteria with 0.1 mg/ml FITC (Sigma-Aldrich) in carbonate buffer (pH 9.5) for 20 min at room-

temperature (RT). 4x10$^6$ shCtrl and shVCL THP1 cells were differentiated in 10 cm$^2$ petri dishes and infected with FITC-labeled *P. gingivalis* at MOI 100 for 1.5 h. Cells were extensively washed and then collected using 5mM ethylenediaminetetraacetic acid (EDTA) (pH 7.5) at 4˚C for 1 h, followed by fixation with 4% PFA in fluorescence-activated cell sorting (FACS) buffer containing 5mM EDTA. FITC-positive cells were analyzed by using a BD Accuri C6 Plus Flow Cytometer.

## Computational docking and analysis of TLR2-VCL interaction

TLR2-VCL interaction was studied using three computational approaches: pyDockEneRes, Optimal Docking Area (ODA) [21] and NIPS (Normalized Interface Propensity) [53], modules from the pyDock docking protein-protein package [23] which predict and evaluate the geometry and energetic profile of protein-protein interactions. Molecular graphics and analyses were performed with UCSF ChimeraX [54], developed by the Resource for Biocomputing, Visualization, and Informatics at the University of California, San Francisco, with support from National Institutes of Health R01-GM129325 and the Office of Cyber Infrastructure and Computational Biology, National Institute of Allergy and Infectious Diseases.

## Site-Directed mutagenesis

The primers carrying phosphorylated 5' mutant nucleotide used to create W>A684 and F>A719 are listed in S2 Table. PCR based site-directed mutagenesis (SDM) was performed with back-to-back primers on bait-TLR2-Cub-V5 and pcDNA3-TLR2-YFP. PCR induced nick was sealed with a T4 DNA ligase enzyme (NEB, MA, USA) and transformed into *E. coli* DH5α (NEB). Plasmids were isolated from the bacteria and the mutations were sequence confirmed. Single mutant bait-TLR2-Cub-V5 and pcDNA3-TLR2-YFP plasmids were used as a PCR template for creating double mutations (AA-TLR2), and plasmids were sequence confirmed.

## Reporter Luciferase assay

Stably transduced NF-κB-HEK293 reporter cells were seeded on 24 well plates and transiently transfected using polyethylenimine (PEI) (Polysciences, PA, USA) at DNA to PEI ratio of 1:2. Cells were transfected with YFP fusion proteins of native TLR2 (WF), or the mutants of TLR2 (W684A, F719A, or W684A/F719A double mutant) together with overexpression of TLR1 and MyD88. After 18 h, transfected cells were stimulated with PAM or left untreated for 4 h followed by lysing with cell-culture lysis reagent (Promega, WI, USA). Luciferase activity was measured with Bright-Glo Luciferase Assay System (Promega) on a GENios Microplate Reader (Tecan) and data were calculated as luciferase activity (relative luminescence units) relative to control cells transfected in an identical manner but not activated with PAM.

## Co-immunofluorescence

Briefly, HeLa cells were seeded at 3x10$^4$ in μ-Slide 8 well IbiDi chambers (ibidi, GmbH, Germany) and transfected using PEI (Polysciences) at DNA to PEI ratio of 1:2. The transfection cocktail was mixed well and then incubated for 15–20 min at room temperature and added dropwise on the cells and incubated for 18 h. Cells transfected with pcDNA3-TLR2-YFP were exposed to *P. gingivalis* at MOI 10 for 30 min and then fixed with 2% PFA for 15 min, followed by blocking and staining with 1μg/ml rabbit anti-VCL (Sigma-Aldrich) followed by counter staining with polyclonal Cy3-labeled anti-rabbit antibody (Jackson ImmunoResearch, PA, USA). Nuclei were counter-stained with Hoechst 33342 (Santa Cruz Biotechnology, TX,

USA). Images were obtained by the aid of a confocal NIKON Eclipse Ti microscope at 40X and 60X (NIKON Instruments Inc, NY, USA). To study co-immunofluorescence in THP1 cells, THP1 monocytes were differentiated with PMA for 72 h and then stimulated and fixed as for the HeLa cells. TLR2 was detected using anti-TLR2 mAb clone T2.5 (Hycult) followed by Alexaflour 488-linked goat anti-mouse secondary antibody (Jackson ImmunoResearch), and VCL was detected as above. For PIP2-VCL co-immunofluorescence THP1 cells were fixed in 4% PFA/0.05% GTA and were blocked with 10 µg/ml digitonin, as in [55]. PIP2 was detected using mouse anti-PIP2 (Abcam) and Alexaflour 488 goat anti-mouse (Jackson ImmunoResearch). The colocalization events were quantified by the JACoP toolbox under NIH ImageJ using Pearson and Manders' coefficient correlation [20]. Six to ten fields including a minimum of 80 cells per group were analyzed.

## Bait and Prey vector cloning for MaMTH

Primers used to generate bait and prey vectors are listed in S2 Table. All cloning was done using the Gateway Cloning System according to the manufacturer's instructions (Invitrogen, CA, USA). Sequences were confirmed by Sanger sequencing (Hylabs, Israel). $6 \times 10^5$ MaMTH-modified HEK293T were seeded on 6-well plates and transfected with bait-TLR2 along with prey vectors using PEI (Polysciences) at DNA to PEI ratio of 1:2. The transfection cocktail was mixed, incubated for 15–20 min at room temperature, and added dropwise to the cells. After 18–20 h cells were stimulated by addition of PAM (10 ng/mL) or infection with *P. gingivalis* MOI 10 for 4, 6, and 8 h at which point cells were lysed using 1X cell-culture lysis reagent (Promega). All repeat experiments were performed at 8 h which was the time-point of maximal signal. Luciferase activity was measured with Bright-Glo™ Luciferase Assay System (Promega) on a GENios Microplate Reader (Tecan).

## Statistical analysis

All analyses were performed using Prism v.8 software (GraphPad Software Inc. San Diego, USA). Two-way analyses were performed by 2-Tailed t-test and multiple comparisons by one-way ANOVA. Values are shown for data that reached a significance of $P \geq 0.05$ (ns), $P \leq 0.05$ (*), $P \leq 0.01$ (**), $P \leq 0.005$ (***), $P \leq 0.0001$ (****). Bars show mean and standard deviation (s.d).

## Supporting information

**S1 Fig. Immunoprecipitation of TLR2 with and without cross-linking. (A)** Differentiated THP1 cells were stimulated with *P. gingivalis* (MOI 10) for the indicated times followed by immunoprecipitation of TLR2. Eluates were analyzed for VCL and TLR2, and whole cell lysates (WCL) were analyzed for VCL. **(B)** Differentiated PBMCs were infected with *P. gingivalis* (MOI 10) for 30 min followed by cross-linking with DSP (M, marker). TLR2 was immunoprecipitated (IP) and the boiled beads were analyzed for VCL by immunoblot (IB). Whole cell lysates (WCL) were analyzed to control for protein input and loading.
(TIF)

**S2 Fig. TLR2 TIR domain-VCL in-silico docking.** Five docking poses obtained from the pyDock docking simulation are shown in distinct colors. Residues W684 and F719 are shown in space-filling model.
(TIF)

**S3 Fig. Membrane localization of TLR2 site-directed mutant constructs.** HEK293 cells were transiently transfected with native TLR2 (WF) and mutants W684A, F719A and

AA-TLR2-YFP (green) and cells were fixed and nuclei were stained with Hoechst (blue). Images were captured using a NIKON confocal microscope at 60X magnification.
(TIF)

**S4 Fig. Expression of bait and prey constructs. (A)** MaMTH-modified HEK293T cells were transfected with respective Nub-prey vectors, and lysed after 48 h. Western blot shows the expression of prey constructs detected using anti-flag antibody. **(B)** HEK293 cells were transiently transfected with TLR2-Cub-V5 and mutants W684A, F719A and AA-TLR2-Cub-V5 (green) and cells were fixed and stained for nuclei with Hoechst (blue). Images were captured using a NIKON confocal microscope at 40X magnification.
(TIF)

**S1 Table. List of cross-linked proteins identified by MS.** TLR2-immunoprecipitated proteins identified by MS containing the CAM modification in resting and *P. gingivalis*-infected THP1 cells.
(XLSX)

**S2 Table. List of the primers used in this manuscript.** Primers used for the RT-PCR, site-directed mutagenesis and Gateway cloning.
(XLSX)

## Acknowledgments

The authors gratefully acknowledge Prof. Asaf Wilensky (Hadassah Medical Center, Jerusalem, Israel) and Prof. Simon Yona (Hebrew University of Jerusalem) for their help with obtaining Helsinki approval for the use of PBMC of healthy volunteers.

## Author Contributions

**Conceptualization:** Karthikeyan Pandi, Jeba Gnanasekaran, Fabian Glaser, Gabriel Nussbaum.

**Formal analysis:** Karthikeyan Pandi, Sarah Angabo, Jeba Gnanasekaran, Hasnaa Makkawi, Luba Eli-Berchoer, Fabian Glaser.

**Funding acquisition:** Gabriel Nussbaum.

**Investigation:** Karthikeyan Pandi, Sarah Angabo, Jeba Gnanasekaran, Hasnaa Makkawi, Luba Eli-Berchoer, Fabian Glaser.

**Methodology:** Karthikeyan Pandi, Sarah Angabo, Jeba Gnanasekaran, Fabian Glaser, Gabriel Nussbaum.

**Project administration:** Gabriel Nussbaum.

**Supervision:** Gabriel Nussbaum.

**Visualization:** Karthikeyan Pandi, Gabriel Nussbaum.

**Writing – original draft:** Karthikeyan Pandi, Fabian Glaser, Gabriel Nussbaum.

**Writing – review & editing:** Gabriel Nussbaum.

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
