## [Decision Letter · Decision Letter 0]

2 Sep 2022

Dear Dr. Nussbaum,

Thank you very much for submitting your manuscript "Porphyromonas gingivalis induction of TLR2 association with Vinculin enables PI3K activation and immune evasion" for consideration at PLOS Pathogens. As with all papers reviewed by the journal, your manuscript was reviewed by members of the editorial board and by several independent reviewers. The reviewers appreciated the impact of your findings and appreciated the well-written manuscript.  They brought attention to a few experiments that would strengthen the study.  In light of the reviews (below this email), we would like to invite the resubmission of a significantly-revised version that takes into account the reviewers' comments.

Please address these major points:

(1) are findings from THP-1 cells recapitulated in primary human monocytes/macrophages?

(2) does paxillin contribute to the TLR2-vinculin interaction?

(3) provide clarification and rationale for statistical tests used.

We cannot make any decision about publication until we have seen the revised manuscript and your response to the reviewers' comments. Your revised manuscript is also likely to be sent to reviewers for further evaluation.

Sincerely,

Mary X. O'Riordan

Associate Editor

PLOS Pathogens

Renée Tsolis

Section Editor

PLOS Pathogens

Kasturi Haldar

Editor-in-Chief

PLOS Pathogens

orcid.org/0000-0001-5065-158X

Michael Malim

Editor-in-Chief

PLOS Pathogens

orcid.org/0000-0002-7699-2064

Please address these major points:

(1) are findings from THP-1 cells recapitulated in primary human monocytes/macrophages?

(2) does paxillin contribute to the TLR2-vinculin interaction

(3) provide clarification and rationale for statistical tests used.

Reviewer's Responses to Questions

**Part I - Summary**

Reviewer #1: The manuscript “Porphyromonas gingivalis induction of THR2 association with vinculin enables PI3K activation and immune evasion” by Pandi et al. presents convincing data that the TLR2-Vanculin protein-protein interaction contributes to P. gingivalis evasion of the immune bactericidal response in macrophages by enabling PIK3/Akt activation. In this very well designed and perfectly executed research the authors show that in vitro upon P. gingivalis uptake by macrophages TLR2 physically interacts with the cytoskeletal vinculin (VCL), what was confirmed using a split-ubiquitin system. Furthermore, residues within the interface essential for this interaction on were determined by mutagenesis. Together, the mechanistic depiction of the novel P. gingivalis strategy to avoid killing by macrophages emerged from this study significantly contributing to our knowledge on the immune evasion by P. gingivalis. This has implications not only to P. gingivalis but also other bacterial pathogens.

Reviewer #2: In this study, Pandi and co-workers described the role of vinculin in TLR2-mediated activation of cells by Porphyromonas gingivalis. In a set of properly designed experiments, using a broad type of methods, the authors demonstrated that vinculin interacts with TLR2. They precisely identified amino acid residues responsible for this association. Moreover, they showed that activation of such a signaling pathway is beneficial for P. gingivalis as activation of PI3K dampens the bactericidal activity of macrophages and promotes intracellular survival of the pathogen. In general, their discovery of the role of vinculin in the regulation of TLR2 signaling during P. gingivalis infection is of high importance.

Reviewer #3: Pandi and colleagues explore the association of vinculin and TLR2 in the context of Porphyromonas gingivalis (Pg) challenge-elicited cellular activation and pathways to microbial evasion of immunity. Previous work highlights the important interaction of Pg with TLR2 in the context of modulation of host immune response and other clinically relevant factors including oral bone loss. Pg targets degradation of MyD88 as a strategy to avoid bactericidal activity and MyD88 independent signaling that requires PI3K. Vinculin is a member of an array of proteins that serve as intracellular connectors of the cell actin skeleton via integrins to extracellular matrix. As Pg is known to target integrin receptors and that TLR2/1 plays a role in this targeting on macrophages the goal of the project was to understand if vinculin contributes to cellular signaling via TLR2. Overall this is a very well conceptualized manuscript that establishes a novel connection between vinculin and host signaling via TLR2. This is felt to be a significant cross-discipline finding that should have significant impact in not only the oral fields, but infectious disease and immunology fields as well that is backed by strong computational, molecular, genetic, proteomic, AA replacement, and infection modeling. This manuscript is logically put together and flows very well. this reviewer has only a few comments regarding this very strong manuscript submission:

**Part II – Major Issues: Key Experiments Required for Acceptance**

Reviewer #1: The manuscript is very well written and the pleasure to read. The only critical comments I have are pertinent to using the human macrophage cell line without verification of key results on primary cells. Therefore, I’m asking the authors to provide arguments for not doing that both in the rebuttal letter and in the discussion.

Reviewer #2: (No Response)

Reviewer #3: 1- (Results lines 88-118). The authors indicate the TLR2 was pulled down in the DSP studies. It is not clear if TLR1 or TLR6 peptides were detected. In a similar light, it does not appear that TLR adaptor molecule peptides were detected either. Could the authors elaborate on this as it would be anticipated that one of these two TLRs would be present as TLR2 signals as a heterodimer as well as the absence of an associating adaptor molecule.

2- (Results lines 88-118). In this same section, STRING analysis predicted that vinculin interaction with TLR2 is possible by is indirect via an interaction with paxlilin. The manuscript does a superb job exploring the TLR-vinculin interaction, but does not clarify experimentally the mechanism of direct vs indirect interaction – in other words is paxillin required for the TLR2-vinculin interaction to occur?

3- The team does excellent work with THP-1 cells which are an established model cell used for exploring macrophage function. Is there any evidence that primary human macrophages recapitulate the findings with the THP-1 cells?

4- In Figure 5D. Was an irrelevant isotype-matched Ab used as a control?

5- Statistical analyses. In some of the assessments made by the authors, there appears to be 3-way comparisons; however 2-way t-tests are indicated as the sole way data were analyzed. Unclear why ANOVA analyses were not considered?

**Part III – Minor Issues: Editorial and Data Presentation Modifications**

Reviewer #1: 1. Abbreviations are spelled out in the result section but not in the Methods section. This makes the method section difficult to comprehend if reading independently.

2. Page 17, lane 369: Apart from “1% 1M HEPES” it is appropriate to specify the final HEPES concentration (in mM). What was sodium pyruvate concentration?

3. Page 18, lane 384: Is there a reference to statement that OD650 of 0.1 corresponds to 10^10^ CFU/mL?

4. Page 20, lane 430: at what MOI?

Reviewer #2: 1. Line 59-60 – there are others receptors evolved to recognize P. gingivalis. They should be listed.

2. Line 97 – the corresponding Figure/Table should be indicated

3. Line 118 – lack of information about PAM stimulation (Fig. 1C)

4. Line 153 – „HeLa cells, widely used for VCL studies …” – this sentence should appear already in the description of Fig. 1.

5. Figure 3A – the control panel showing transfected cells without PAM stimulation should be presented.

6. Line 159 – „ …TLR2 co-localization was not significantly different …” – „ns” should be indicated in the Fig. 3B

7. The lack of statistical analysis for PAM stimulation between mutants in the Fig. 4E. Moreover, the phenomenon was described for P. gingivalis but as PAM lipopeptide stimulation can lead to a similar effect (Fig. 4) the role of such signaling pathway for other pathogens should be discussed – at least the activation of PI3K by PAM and its immunomodulatory role.

8. Fig. 4D – could be presented as supplementary data

9. Line 251 – „P. gingivalis blocks phagolysosomal maturation and intracellular survival in macrophages…” – the sentence should be rephrased

10. As the authors showed diminished phagocytosis of P. gingivalis in VCL knock-down macrophages (Fig. 5E) the percent of recovery could be calculated based on the comparison of engulfed bacteria (after 1 h of phagocytosis) to the number of CFU obtained after different time post infection. Those data could be presented as a separate panel. What is the MOI used in the analysis of bacterial survival? Was the analysis of bacterial recovery performed after longer time post infection (days)?

11. Figure 7 – what do the authors mean as steady state (1)? Non-infected cells? Then, how the improved killing of bacteria can be proposed?

12. Line 506 – lack of description for „****”

13. Line 657 – the figure legend should be supplemented by adding a description of **, ***, ns

14. Line 690 – the activation of TF was performed after 8h. Is it the optimal time for measurement of its activity?

15. Line 717 – lack of description for „****”

16. Supplementary Table 1 – it would be valuable to present the data in a form that allows for the straight comparison of protein identified in resting and P. gingivalis-infected cells.

Reviewer #3: (No Response)

PLOS authors have the option to publish the peer review history of their article (what does this mean?). If published, this will include your full peer review and any attached files.

Reviewer #1: No

Reviewer #2: **Yes: **Joanna Koziel

Reviewer #3: No
---

## [Editor Report · Decision Letter 1]

9 Mar 2023

Dear Dr. Nussbaum,

We are pleased to inform you that your manuscript 'Porphyromonas gingivalis induction of TLR2 association with Vinculin enables PI3K activation and immune evasion' has been provisionally accepted for publication in PLOS Pathogens.

Best regards,

Renée Tsolis

Section Editor

PLOS Pathogens

Kasturi Haldar

Editor-in-Chief

PLOS Pathogens

orcid.org/0000-0001-5065-158X

Michael Malim

Editor-in-Chief

PLOS Pathogens

orcid.org/0000-0002-7699-2064

---

## [Editor Report · Acceptance letter]

2 Apr 2023

Dear Dr. Nussbaum,

We are delighted to inform you that your manuscript, " Porphyromonas gingivalis induction of TLR2 association with Vinculin enables PI3K activation and immune evasion," has been formally accepted for publication in PLOS Pathogens.

Best regards,

Kasturi Haldar

Editor-in-Chief

PLOS Pathogens

orcid.org/0000-0001-5065-158X

Michael Malim

Editor-in-Chief

PLOS Pathogens

orcid.org/0000-0002-7699-2064